# Intrinsic Dimension, Persistent Homology and Generalization in Neural Networks

**Tolga Birdal**
Stanford University
tbirdal@stanford.edu

**Aaron Lou**
Stanford University
aaronlou@stanford.edu

**Leonidas Guibas**
Stanford University
guibas@cs.stanford.edu

**Umut Şimşekli**
INRIA & ENS – PSL Research University
umut.simsekli@inria.fr

## Abstract

Disobeying the classical wisdom of statistical learning theory, modern deep neural networks generalize well even though they typically contain millions of parameters. Recently, it has been shown that the trajectories of iterative optimization algorithms can possess *fractal structures*, and their generalization error can be formally linked to the complexity of such fractals. This complexity is measured by the fractal's *intrinsic dimension*, a quantity usually much smaller than the number of parameters in the network. Even though this perspective provides an explanation for why overparametrized networks would not overfit, computing the intrinsic dimension (*e.g.*, for monitoring generalization during training) is a notoriously difficult task, where existing methods typically fail even in moderate ambient dimensions. In this study, we consider this problem from the lens of topological data analysis (TDA) and develop a generic computational tool that is built on rigorous mathematical foundations. By making a novel connection between learning theory and TDA, we first illustrate that the generalization error can be equivalently bounded in terms of a notion called the 'persistent homology dimension' (PHD), where, compared with prior work, our approach does not require any additional geometrical or statistical assumptions on the training dynamics. Then, by utilizing recently established theoretical results and TDA tools, we develop an efficient algorithm to estimate PHD in the scale of modern deep neural networks and further provide visualization tools to help understand generalization in deep learning. Our experiments show that the proposed approach can efficiently compute a network's intrinsic dimension in a variety of settings, which is predictive of the generalization error.

## 1 Introduction

In recent years, deep neural networks (DNNs) have become the de facto machine learning tool and have revolutionized a variety of fields such as natural language processing [DCLT18], image perception [KSH12, RBH+21], geometry processing [QSMG17, ZBL+20] and 3D vision [DBI18, GLW+21]. Despite their widespread use, little is known about their theoretical properties. Even now the top-performing DNNs are designed by trial-and-error, a pesky, burdensome process for the average practitioner [EMH+19]. Furthermore, even if a top-performing architecture is found, it is difficult to provide performance guarantees on a large class of real-world datasets.

This lack of theoretical understanding has motivated a plethora of work focusing on explaining what, how, and why a neural network learns. To answer many of these questions, one naturally examines the generalization error, a measure quantifying the differing performance on train and

35th Conference on Neural Information Processing Systems (NeurIPS 2021).

test data since this provides significant insights into whether the network is learning or simply memorizing [ZBH+21]. However, generalization in neural networks is particularly confusing as it refutes the classical proposals of statistical learning theory such as uniform bounds based on the Rademacher complexity [BM02] and the Vapnik–Chervonenkis (VC) dimension [Vap68].

Instead, recent analyses have started focusing on the dynamics of deep neural networks. [NBMS17, BO18, GJ16] provide analyses on the final trained network, but these miss out on critical training patterns. To remedy this, a recent study [SSDE20] connected generalization and the heavy tailed behavior of *network trajectories*–a phenomenon which had already been observed in practice [SSG19, ŞGN+19, SZTG20, GSZ21, CWZ+21, HM20, MM19]. [SSDE20] further showed that the generalization error can be linked to the *fractal dimension* of a parametric hypothesis class (which can then be taken as the optimization trajectories). Hence, the fractal dimension acts as a 'capacity metric' for generalization.

While [SSDE20] brought a new perspective to generalization, several shortcomings prevent application in everyday training. In particular, their construction requires several conditions which may be infeasible in practice: (i) topological regularity conditions on the hypothesis class for fast computation, (ii) a Feller process assumption on the training algorithm trajectory, and that (iii) the Feller process exhibits a specific diffusive behavior near a minimum. Furthermore, the capacity metrics in [SSDE20] are not optimization friendly and therefore can't be incorporated into training.

In this work, we address these shortcomings by exploiting the recently developed connections between fractal dimension and topological data analysis (TDA). First, by relating the *box dimension* [Sch09] and the recently proposed *persistent homology (PH) dimension* [Sch20], we relax the assumptions in [SSDE20] to develop a topological intrinsic dimension (ID) estimator. Then, using this estimator we develop a general tool for *computing* and *visualizing* generalization properties in deep learning. Finally, by leveraging recently developed differentiable TDA tools [CHU17, CHN19], we employ our ID estimator to regularize training towards solutions that generalize better, even without having access to the test dataset.

Our experiments demonstrate that this new measure of intrinsic dimension correlates highly with generalization error, regardless of the choice of optimizer. Furthermore, as a proof of concept, we illustrate that our topological regularizer is able to improve the test accuracy and lower the generalization error. In particular, this improvement is most pronounced when the learning rate/batch size normally results in a poorer test accuracy.

Overall, our contributions are summarized as follows:

- We make a novel connection between statistical learning theory and TDA in order to develop a generic computational framework for the generalization error. We remove the topological regularity condition and the decomposable Feller assumption on training trajectories, which were required in [SSDE20]. This leads to a more generic capacity metric.

- Using insights from our above methodology, we leverage the differentiable properties of persistent homology to regularize neural network training. Our findings also provide the first steps towards theoretically justifying recent topological regularization methods [BGND+19, CNBW19].

- We provide extensive experiments to illustrate the theory, strength, and flexibility of our framework.

We believe that the novel connections and the developed framework will open new theoretical and computational directions in the theory of deep learning. To foster further developments at the the intersection of persistent homology and statistical learning theory, we release our source code under: https://github.com/tolgabirdal/PHDimGeneralization.

## 2 Related Work

**Intrinsic dimension in deep networks**    Even though a large number of parameters are required to train deep networks [FC18], modern interpretations of deep networks avoid correlating model over-fitting or generalization to parameter counting. Instead, contemporary studies measure model complexity through the degrees of freedom of the parameter space [JFH15, GJ16], compressibility (pruning) [BO18] or intrinsic dimension [ALMZ19, LFLY18, MWH+18]. Tightly related to the ID, Janson *et al*. [JFH15] investigated the *degrees of freedom* [Ghr10] in deep networks and expected difference between test error and training error. Finally, LDMNet [ZQH+18] explicitly penalizes the ID regularizing the network training.

**Generalization bounds**  Several studies have provided theoretical justification to the observations that trained neural networks live in a lower-dimensional space, and this is related to the generalization performance. In particular, compression-based generalization bounds [AGNZ18, SAM$^+$20, SAN20, HJTW21, BSE$^+$21] have shown that the generalization error of a neural network can be much lower if it can be accurately represented in lower dimensional space. Approaching the problem from a geometric viewpoint, [SSDE20] showed that the generalization error can be formally linked to the fractal dimension of a parametric hypothesis class. This dimension indeed the plays role of the intrinsic dimension, which can be much smaller than the ambient dimension. When the hypothesis class is chosen as the trajectories of the training algorithm, [SSDE20] further showed that the error can be linked to the heavy-tail behavior of the trajectories.

**Deep networks & topology**  Previous works have linked neural network training and topological invariants, although all analyze the final trained network [FGFAEV21]. For example, in [RTB$^+$19], the authors construct Neural Persistence, a measure on neural network layer weights. They furthermore show that Neural Persistence reflects many of the properties of convergence and can classify weights based on whether they overfit, underfit, or exactly fit the data. In a parallel line of work, [DZF19] analyze neural network training by calculating topological properties of the underlying graph structure. This is expanded upon in [CMEM20], where the authors compute correlations between neural network weights and show that the homology is linked with the generalization error.

However, these previous constructions have been done mostly in an adhoc manner. As a result, many of the results are mostly empirical and work must still be done to show that these methods hold theoretically. Our proposed method, by contrast, is theoretically well-motivated and uses tools from statistical persistent homology theory to formally links the generalization error with the network training trajectory topology.

We also would like to note that prior work has incorporated topological loss functions to help normalize training. In particular, [BGND$^+$19] constructed a topological normalization term for GANs to help maintain the geometry of the generated 3d point clouds.

## 3   Preliminaries & Technical Background

We imagine a point cloud $W = \{w_i \in \mathbb{R}^d\}$ as a geometric realization of a $d$-dimensional topological space $W \subset \mathcal{W} \subset \mathbb{R}^d$. $B_\delta(x) \subset \mathbb{R}^d$ denotes the closed ball centered around $x \in \mathbb{R}^d$ with radius $\delta$.

**Persistent Homology**  From a topological perspective, $\mathcal{W}$ can be viewed a *cell complex* composed of the disjoint union of $k$-dimensional balls or *cells* $\sigma \in \mathcal{W}$ *glued* together. For $k = 0, 1, 2, \ldots$, we form a *chain complex* $\mathcal{C}(\mathcal{W}) = \ldots C_{k+1}(\mathcal{W}) \xrightarrow{\partial_{k+1}} C_k(\mathcal{W}) \xrightarrow{\partial_k} \ldots$ by sequencing *chain groups* $C_k(\mathcal{W})$, whose elements are equivalence classes of cycles, via boundary maps $\partial_k : C_k(\mathcal{W}) \mapsto C_{k-1}(\mathcal{W})$ with $\partial_{k-1} \circ \partial_k \equiv 0$. In this paper, we work with finite *simplicial complexes* restricting the cells to be simplices.

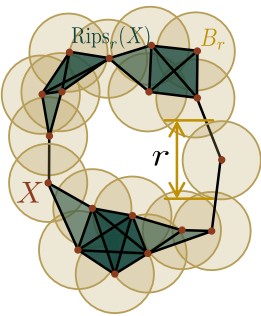

The $k^{\text{th}}$ homology group or *k-dimensional homology* is then defined as the equivalence classes of $k$-dimensional cycles who differ only by a boundary, or in other words, the quotient group $H_k(\mathcal{W}) = Z_k(\mathcal{W})/Y_k(\mathcal{W})$ where $Z_k(\mathcal{W}) = \ker \partial_k$ and $Y_k(\mathcal{W}) = \operatorname{im} \partial_{k+1}$. The generators or *basis* of $H_0(\mathcal{W}), H_1(\mathcal{W})$ and $H_2(\mathcal{W})$ describe the shape of the topological space $\mathcal{W}$ by its connected components, holes and cavities, respectively. Their ranks are related to the *Betti numbers i.e.* $\beta_k = \operatorname{rank}(H_k)$.

Figure 1: A visualization of a Vietoris-Rips complex computed using persistent homology (PH).

**Definition 1** (Čech and Vietoris-Rips Complexes)**.** *For $W$ a set of fine points in a metric space, the Čech cell complex $\check{C}ech_r(W)$ is constructed using the intersection of $r$-balls around $W$, $B_r(W)$: $\check{C}ech_r(W) = \{Q \subset W : \cap_{x \in \mathbf{Q}} B_r(x) \neq 0\}$. The construction of such complex is intricate. Instead, the Vietoris-Rips complex $VR_r(W)$ closely approximates $\check{C}ech_r(W)$ using only the pairwise distances or the intersection of two $r$-balls [RB21]:* $\mathcal{W}_r = VR_r(W) = \{Q \subset W : \forall x, x' \in Q, B_r(x) \cap B_r(x') \neq 0\}$.

**Definition 2** (Persistent Homology). *PH indicates a multi-scale version of homology applied over a filtration* $\{\mathcal{W}_t\}_t := VR(W) : \forall (s \leq t)\, \mathcal{W}_s \subset \mathcal{W}_t \subset \mathcal{W}$, *keeping track of holes created (born) or filled (died) as $t$ increases. Each persistence module* $\mathrm{PH}_k(VR(W)) = \{\gamma_i\}_i$ *keeps track of a single $k$-persistence cycle $\gamma_i$ from* birth *to* death. *We denote the entire lifetime of cycle $\gamma$ as $I(\gamma)$ and its length as* $|I(\gamma)| = \mathrm{death}(\gamma) - \mathrm{birth}(\gamma)$. *We will also use* persistence diagrams, *2D plots of all persistence lifetimes (death vs. birth). Note that for* $\mathrm{PH}_0$, *the Čech and VR complexes are equivalent.*

Lifetime intervals are instrumental in TDA as they allow for extraction of topological features or summaries. Note that, each birth-death pair can be mapped to the cells that respectively created and destroyed the homology class, defining a unique map for a persistence diagram, which lends itself to *differentibility* [BGND+19, CHN19, CHU17]. We conclude this brief section by referring the interested reader to the well established literature of persistent homology [Car14, EH10] for a thorough understanding.

**Intrinsic Dimension**    The intrinsic dimension of a space can be measured by using various notions. In this study, we will consider two notions of dimension, namely the upper-box dimension (also called the Minkowski dimension) and the persistent homology dimension. The box dimension is based on covering numbers and can be linked to generalization via [SSDE20], whereas the PH dimension is based on the notions defined earlier in this section.

We start by the box dimension.

**Definition 3** (Upper-Box Dimension). *For a **bounded metric space** $\mathcal{W}$, let $N_\delta(\mathcal{W})$ denote the maximal number of disjoint closed $\delta$-balls with centers in $\mathcal{W}$. The* upper box dimension *is defined as:*

$$\dim_{\mathrm{Box}} \mathcal{W} = \limsup_{\delta \to 0} \Big( \log(N_\delta(\mathcal{W}))/\log(1/\delta) \Big). \tag{1}$$

We proceed with the PH dimension. First let us define an intermediate construct, which will play a key role in our computational tools.

**Definition 4** ($\alpha$-Weighted Lifetime Sum). *For a **finite set** $W \subset \mathcal{W} \subset \mathbb{R}^d$, the weighted $i^{\mathrm{th}}$ homology lifetime sum is defined as follows:*

$$E_\alpha^i(W) = \sum_{\gamma \in \mathrm{PH}_i(\mathrm{VR}(W))} |I(\gamma)|^\alpha, \tag{2}$$

*where* $\mathrm{PH}_i(\mathrm{VR}(W))$ *is the $i$-dimensional persistent homology of the Čech complex on a finite point set $W$ contained in $\mathcal{W}$ and $|I(\gamma)|$ is the persistence lifetime as explained above.*

Now, we are ready to define the PH dimension, which is the key notion in this paper.

**Definition 5** (Persistent Homology Dimension). *The $\mathrm{PH}_i$-dimension of a **bounded metric space** $\mathcal{W}$ is defined as follows:*

$$\dim_{\mathrm{PH}}^i \mathcal{W} := \inf \big\{ \alpha \, : \, E_\alpha^i(W) < C; \quad \exists C > 0, \forall \textit{ finite } W \subset \mathcal{W} \big\}. \tag{3}$$

In words, $\dim_{\mathrm{PH}}^i \mathcal{W}$ is the smallest exponent $\alpha$ for which $E_\alpha^i$ is uniformly bounded for all finite subsets of $\mathcal{W}$.

## 4    Generalization Error via Persistent Homology Dimension

In this section, we will illustrate that the generalization error can be linked to the $\mathrm{PH}_0$ dimension. Our approach is based on the following fundamental result.

**Theorem 1** ([KLS06, Sch19]). *Let $\mathcal{W} \subset \mathbb{R}^d$ be a bounded set. Then, it holds that:*

$$\dim_{\mathrm{PH}} \mathcal{W} := \dim_{\mathrm{PH}}^0 \mathcal{W} = \dim_{\mathrm{Box}} \mathcal{W}.$$

In the light of this theorem, we combine the recent result showing that the generalization error can be linked to the box dimension [SSDE20], and Theorem 1, which shows that, for bounded subsets of $\mathbb{R}^d$, the box dimension and the PH dimension of order 0 agree.

By following the notation of [SSDE20], we consider a standard supervised learning setting, where the data space is denoted by $\mathcal{Z} = \mathcal{X} \times \mathcal{Y}$, and $\mathcal{X}$ and $\mathcal{Y}$ respectively denote the features and the labels. We assume that the data is generated via an unknown data distribution $\mathcal{D}$ and we have access to a training set of $n$ points, i.e., $S = \{z_1, \ldots, z_n\}$, with the samples $\{z_i\}_{i=1}^n$ are independent and identically (i.i.d.) drawn from $\mathcal{D}$.

We further consider a parametric hypothesis class $\mathcal{W} \subset \mathbb{R}^d$, that potentially depends on $S$. We choose $\mathcal{W}$ to be *optimization trajectories* given by a training algorithm $\mathcal{A}$, which returns the entire (random) trajectory of the network weights in the time frame $[0, T]$, such that $[\mathcal{A}(S)]_t = w_t$ being the network weights returned by $\mathcal{A}$ at 'time' $t$, and $t$ is a continuous iteration index. Then, in the set $\mathcal{W}$, we collect all the network weights that appear in the optimization trajectory:

$$\mathcal{W} := \{w \in \mathbb{R}^d : \exists t \in [0, T], w = [\mathcal{A}(S)]_t\}$$

where we will set $T = 1$, without loss of generality.

To measure the quality of a parameter vector $w \in \mathcal{W}$, we use a loss function $\ell : \mathbb{R}^d \times \mathcal{Z} \mapsto \mathbb{R}_+$, such that $\ell(w, z)$ denotes the loss corresponding to a single data point $z$. We then denote the population and empirical risks respectively by $\mathcal{R}(w) := \mathbb{E}_z[\ell(w, z)]$ and $\hat{\mathcal{R}}(w, S) := \frac{1}{n}\sum_{i=1}^n \ell(w, z_i)$. The generalization error is hence defined as $|\hat{\mathcal{R}}(w, S) - \mathcal{R}(w)|$.

We now recall [SSDE20, Asssumption H4], which is a form of algorithmic stability [BE02]. Let us first introduce the required notation. For any $\delta > 0$, consider the fixed grid on $\mathbb{R}^d$,

$$G = \left\{ \left( \frac{(2j_1 + 1)\delta}{2\sqrt{d}}, \ldots, \frac{(2j_d + 1)\delta}{2\sqrt{d}} \right) : j_i \in \mathbb{Z}, i = 1, \ldots, d \right\},$$

and define the set $N_\delta := \{x \in G : B_\delta(x) \cap \mathcal{W} \neq \emptyset\}$, that is the collection of the centers of each ball that intersect $\mathcal{W}$.

**H1.** *Let $\mathcal{Z}^\infty := (\mathcal{Z} \times \mathcal{Z} \times \cdots)$ denote the countable product endowed with the product topology and let $\mathfrak{B}$ be the Borel $\sigma$-algebra generated by $\mathcal{Z}^\infty$. Let $\mathfrak{F}, \mathfrak{G}$ be the sub-$\sigma$-algebras of $\mathfrak{B}$ generated by the collections of random variables given by $\{\hat{\mathcal{R}}(w, S) : w \in \mathbb{R}^d, n \geq 1\}$ and $\left\{ \mathbb{1}\{w \in N_\delta\} : \delta \in \mathbb{Q}_{>0}, w \in G, n \geq 1 \right\}$ respectively. There exists a constant $M \geq 1$ such that for any $A \in \mathfrak{F}$, $B \in \mathfrak{G}$ we have $\mathbb{P}[A \cap B] \leq M\mathbb{P}[A]\mathbb{P}[B]$.*

The next result forms our main observation, which will lead to our methodological developments.

**Proposition 1.** *Let $\mathcal{W} \subset \mathbb{R}^d$ be a (random) compact set. Assume that **H**1 holds, $\ell$ is bounded by $B$ and $L$-Lipschitz continuous in $w$. Then, for $n$ sufficiently large, we have*

$$\sup_{w \in \mathcal{W}} |\hat{\mathcal{R}}(w, S) - \mathcal{R}(w)| \leq 2B\sqrt{\frac{[\dim_{\mathrm{PH}}\mathcal{W} + 1]\log^2(nL^2)}{n} + \frac{\log(7M/\gamma)}{n}}, \qquad (4)$$

*with probability at least $1 - \gamma$ over $S \sim \mathcal{D}^{\otimes n}$.*

*Proof.* By using the same proof technique as [SSDE20, Theorem 2], we can show that (4) holds with $\dim_{\mathrm{Box}}\mathcal{W}$ in place of $\dim_{\mathrm{PH}}\mathcal{W}$. Since $\mathcal{W}$ is bounded, we have $\dim_{\mathrm{Box}}\mathcal{W} = \dim_{\mathrm{PH}}\mathcal{W}$ by Theorem 1. The result follows. $\qquad \square$

This result shows that the generalization error of the trajectories of a training algorithm is deeply linked to its topological properties as measured by the PH dimension. Thanks to novel connection, we have now access to the rich TDA toolbox, to be used for different purposes.

### 4.1 Analyzing Deep Network Dynamics via Persistent Homology

By exploiting TDA tools, our goal in this section is to develop an algorithm to compute $\dim_{\mathrm{PH}}\mathcal{W}$ for two main purposes. The first goal is to predict the generalization performance by using $\dim_{\mathrm{PH}}$. By this approach, we can use $\dim_{\mathrm{PH}}$ for hyperparameter tuning without having access to test data. The second goal is to incorporate $\dim_{\mathrm{PH}}$ as a regularizer to the optimization problem in order to improve generalization. Note that similar topological regularization strategies have already been proposed

---

**Algorithm 1:** Computation of $\dim_{\mathrm{PH}}$.

**1 input :** The set of iterates $W = \{w_i\}_{i=1}^K$, smallest sample size $n_{\min}$, and a skip step $\Delta$, $\alpha$
**2 output :** $\dim_{\mathrm{PH}} W$
**3** $n \leftarrow n_{\min}, \quad E \leftarrow []$
**4 while** $n \leq K$ **do**
**5** $\quad$ $W_n \leftarrow \text{sample}(W, n)$ // `random sampling`
**6** $\quad$ $\mathcal{W}_n \leftarrow \text{VR}(W_n)$ // `Vietoris-Rips filtration`
**7** $\quad$ $E[i] \leftarrow E_\alpha(\mathcal{W}_n) \triangleq \sum_{\gamma \in \mathrm{PH}_0(\mathcal{W}_n)} |I(\gamma)|^\alpha$ // `compute lifetime sums from PH`
**8** $\quad$ $n \leftarrow n + \Delta$
**9** $m, b \leftarrow \text{fitline}(\log(n_{\min} : \Delta : K), \log(E))$ // `power law on` $E_1^i(W)$
**10** $\dim_{\mathrm{PH}} W \leftarrow \frac{\alpha}{1-m}$

---

[BGND$^+$19, CNBW19] without a formal link to generalization. In this sense, our observations form the first step towards theoretically linking generalization and TDA.

In [SSDE20], to develop a computational approach, the authors first linked the intrinsic dimension to certain statistical properties of the underlying training algorithm, which can be then estimated. To do so, they required an additional topological regularity condition, which necessitates the existence of an 'Ahlfors regular' measure defined on $\mathcal{W}$, *i.e.*, a finite Borel measure $\mu$ such that there exists $s, r_0 > 0$ where $0 < ar^s \leq \mu(B_r(x)) \leq br^s < \infty$, holds for all $x \in \mathcal{W}, 0 < r \leq r_0$. This assumption was used to link the box dimension to another notion called Hausdorff dimension, which can be then linked to statistical properties of the training trajectories under further assumptions (see Section 1). An interesting asset of our approach is that, we do not require this condition and thanks to the following result, we are able to develop an algorithm to directly estimate $\dim_{\mathrm{PH}} \mathcal{W}$, while staying agnostic to the finer topological properties of $\mathcal{W}$.

**Proposition 2.** *Let $\mathcal{W} \subset \mathbb{R}^d$ be a bounded set with $\dim_{\mathrm{PH}} \mathcal{W} =: d^\star$. Then, for all $\varepsilon > 0$ and $\alpha \in (0, d^\star + \varepsilon)$, there exists a constant $D_{\alpha,\varepsilon}$, such that the following inequality holds for all $n \in \mathbb{N}_+$ and all collections $W_n = \{w_1, \ldots, w_n\}$ with $w_i \in \mathcal{W}$, $i = 1, \ldots, n$:*

$$E_\alpha^0(W_n) \leq D_{\alpha,\varepsilon} n^{\frac{d^\star + \varepsilon - \alpha}{d^\star + \varepsilon}}. \tag{5}$$

*Proof.* Since $\mathcal{W}$ is bounded, we have $\dim_{\mathrm{Box}} \mathcal{W} = d^\star$ by Theorem 1. Fix $\varepsilon > 0$. Then, by Definition 3, there exists $\delta_0 = \delta_0(\varepsilon) > 0$ and a finite constant $C_\varepsilon > 0$ such that for all $\delta \leq \delta_0$ the following inequality holds:

$$N_\delta(\mathcal{W}) \leq C_\varepsilon \delta^{-(d^\star + \varepsilon)}. \tag{6}$$

Then, the result directly follows from [Sch20, Proposition 21]. $\qquad\square$

This result suggests a simple strategy to estimate an upper bound of the intrinsic dimension from persistent homology. In particular, we note that rewriting (5) for logarithmic values give us that

$$\left(1 - \frac{\alpha}{d^* + \epsilon}\right) \log n + \log D_{\alpha,\epsilon} \geq \log E_\alpha^0. \tag{7}$$

If $\log E_\alpha^0$ and $\log n$ are sampled from the data and give an empirical slope $m$, then we see that $d^* + \epsilon \leq \frac{m}{1-\alpha}$. In many cases, we see that $d^* \approx \frac{\alpha}{1-m}$ (as further explained in Sec. 5.2), so we take $\frac{\alpha}{1-m}$ as our PH dimension estimation. We provide the full algorithm for computing this from our sampled data in Alg. 1. Note that our algorithm is similar to that proposed in [AAF$^+$20], although our method works for sets rather than probability measures. In our implementation we compute the homology by the celebrated Ripser package [Bau21] unless otherwise specified.

**On computational complexity.** Computing the Vietoris Rips complex is an active area of research, as the worst-case time complexity is meaningless due to natural sparsity [Zom10]. Therefore, to calculate the time complexity of our estimator, we focus on analyzing the PH computation from the output simplices: calculating PH takes $O(p^w)$ time, where $w < 2.4$ is the constant of matrix multiplication and $p$ is the number of simplices produced in the filtration [BP19]. Since we compute

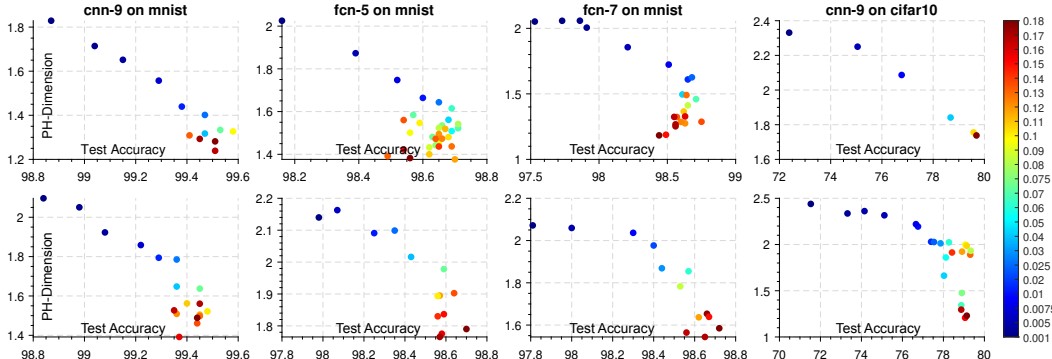

Figure 2: PH-dimension vs test accuracy for different models and datasets. The rows correspond to the model and dataset, and the columns correspond to the batch size (50 and 100 for the top and bottom row respectively). The graphed points are marked with different colors corresponding to the learning rate. Note that the PH dimension is inversely correlated with test accuracy and is thus positively correlated with generalization error.

with $0^{\text{th}}$ order homology, this would imply that the computational complexity is $O(n^w)$, where $n$ is the number of points. In particular, this means that estimating the PH dimension would take $O(kn^w)$ time, where $k$ is the number of samples taken assuming that samples are evenly spaced in $[0, n]$.

## 4.2    Regularizing Deep Networks via Persistent Homology

Motivated by our results in proposition 2, we theorize that controlling $\dim_{\text{PH}} \mathcal{W}$ would help in reducing the generalization error. Towards this end, we develop a regularizer for our training procedure which seeks to minimize $\dim_{\text{PH}} \mathcal{W}$ during train time. If we let $\mathcal{L}$ be our vanilla loss function, then we will instead optimize over our topological loss function $\mathcal{L}_\lambda := \mathcal{L} + \lambda \dim_{\text{PH}} \mathcal{W}$, where $\lambda \geq 0$ controls the scale of the regularization and $\mathcal{W}$ now denotes a sliding window of iterates (*e.g.*, the latest 50 iterates during training). This way, we aim to regularize the loss by considering the dimension of the ongoing training trajectory.

In Alg. 1, we let $w_i$ be the stored weights from previous iterations for $i \in \{1, \ldots, K - 1\}$ and let $w_K$ be the current weight iteration. Since the persistence diagram computation and linear regression are differentiable, this means that our estimate for $\dim_{\text{PH}}$ is also differentiable, and, if $w_k$ is sampled as in Alg. 1, is connected in the computation graph with $w_K$. We incorporate our regularizer into the network training using PyTorch [PGM$^+$19] and the associated persistent homology package *torchph* [CHU17, CHN19].

## 5    Experimental Evaluations

This section presents our experimental results in two parts: (i) analyzing and quantifying generalization in practical deep networks on real data, (ii) ablation studies on a random diffusion process. In all the experiments we will assume that the intrinsic dimension is strictly larger than 1, hence we will set $\alpha = 1$, unless specified otherwise. Further details are reported in the supplementary document.

### 5.1    Analyzing and Visualizing Deep Networks

**Measuring generalization.**    We first verify our main claim by showing that our persistent homology dimension derived from topological analysis of the training trajectories correctly measures of generalization. To demonstrate this, we apply our analysis to a wide variety of networks, training procedures, and hyperparameters. In particular, we train AlexNet [KSH12], a 5-layer (fcn-5) and 7-layer (fcn-7) fully connected networks, and a 9-layer convolutional netowork (cnn-9) on MNIST, CIFAR10 and CIFAR100 datasets for multiple batch sizes and learning rates until convergence. For AlexNet, we consider 1000 iterates prior to convergence and, for the others, we only consider 200. Then, we estimate $\dim_{\text{PH}}$ on the last iterates by using Alg. 1. For varying $n$, we randomly pick $n$ of last iterates and compute $E_\alpha^0$, and then we use the relation given in (5).

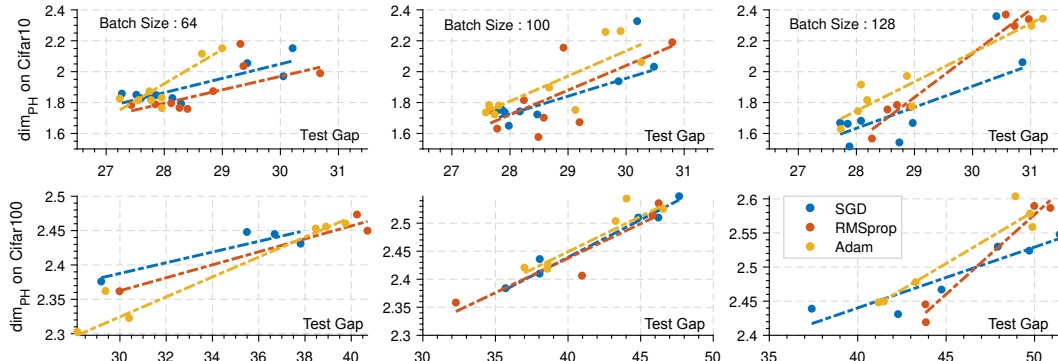

Figure 3: (Estimated) persistent homology dimension vs generalization error (training accuracy - test accuracy) for different datasets (top row CIFAR10, bottom row CIFAR100) and optimizers on AlexNet. We plot the data points and lines of best fit. Note that the PH dimension is *directly* correlated with the generalization error and is consistent across datasets and optimizers.

We obtain the ground truth (GT) generalization error as the gap between training and test accuracies. Fig. 2 plots the PH-dimension with respect to test accuracy and signals a strong correlation of our PH-dimension and actual performance gap. The lower the PH-dimension, the higher the test accuracy. Note that this results aligns well with that of [SSDE20]. The figure also shows that the intrinsic dimensions across different datasets can be similar, even if the parameters of the models can vary greatly. This supports the recent hypothesis that what matters for the generalization is the effective capacity and not the parameter count. In fact, the dimension should be as minimal as possible without collapsing important representation features onto the same dimension. The findings in Fig. 2 are further augmented with results in Fig. 3, where a similar pattern is observed on AlexNet and CIFAR100.

**Can $\dim_{PH}$ capture intrinsic properties of trajectories?** After revealing that our ID estimation is a gauge for generalization, we set out to investigate whether it really hinges on the intrinsic properties of the data. We train several instances of 7-fcn for different learning rates and batch sizes. We compute the PH-dimension of each network using training trajectories. We visualize the following in the rows of Fig. 4 sorted by $\dim_{PH}$: (i) $200 \times 200$ distance matrix of the sequence of iterates $w_1, \ldots, w_K$ (which is the basis for PH computations), (ii) corresponding $\log E_{\alpha=1}^0$ estimates as we sweep over $n$ in an increasing fashion, (iii) persistence diagrams per each distance matrix. It is clear that there is a strong correlation between $\dim_{PH}$ and the structure of the distance matrix. As dimension increases, matrix of distances become non-uniformly *pixelated*. The slope estimated from the total edge lengths the second row is a quantity proportional to our dimension. Note that the slope decreases as our estimte increases (hence generalization tends to decrease). We further observe clusters emerging in the persistence diagram. The latter has also been reported for better generalizing networks, though using a different notion of a topological space [BGND+19].

**Is $\dim_{PH}$ a real indicator of generalization?** To quantitatively assess the quality of our complexity measure, we gather two statistics: (i) we report the average $p$-value over different batch sizes for AlexNet trained with SGD on the Cifar100 dataset. The value of $p = 0.0157 < 0.05$ confirms the statistical significance. Next, we follow the recent literature [JFY+20] and consult the Kendall correlation coefficient (KCC). Similar to the $p$-value experiment above, we compute KCC for AlexNet+SGD for different batch sizes $(64, 100, 128)$ and attain $(0.933, 0.357, 0.733)$ respectively. Note that, a positive correlation signals that the test gap closes as $\dim_{PH}$ decreases. Both of these experiments agree with our theoretical insights that connect generalization to a topological characteristic of a neural network: intrinsic dimension of training trajectories.

**Effect of different training algorithms.** We also verify that our method is algorithm-agnostic and does not require assumptions on the training algorithm. In particular, we show that our above analyses extend to both the RMSProp [TH12] and Adam [KB15] optimizer. Our results are visualized in Fig. 3. We plot the dimension with respect to the generalization error for varying optimizers and batch sizes; our results verify that the generalization error (which is inversely related to the test accuracy) is

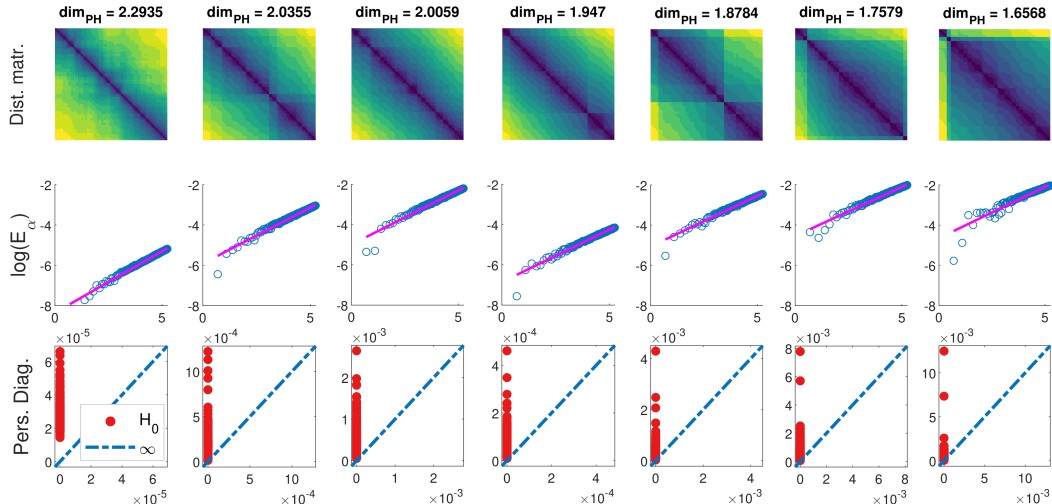

Figure 4: We visualize topological information for a 7-layer fully connected network on CIFAR10 data. In the top row, we visualize the distance matrices computed between network weights corresponding to the last 200 iterations of training. In the middle, we visualize the corresponding behavior of our estimator as we increase the number of samples. In the bottom row, we visualize the 0-th order persistent diagrams for the full data. As our PH dimension decreases, the matrix becomes more segmented, the estimator slope decreases, and the persistent diagram becomes sparser. We provide more information about these results in the supplement.

positively correlated with the PH dimension. This corroborates our previous results in Fig. 2 and in particular shows that our dimension estimator of test gap is indeed algorithm-agnostic.

**Encouraging generalization via regularization** $\dim_{\mathrm{PH}}$. We furthermore verify that our topological regularizer is able to help control the test gap in accordance with our theory. We train a Lenet-5 network [LBBH98] on Cifar10 [Kri09] and compare a clean trianing with a training with our topological regularizer with $\lambda$ set to $1$. We train for $200$ epochs with a batch size of $128$ and report the train and test accuracies in Fig. 5 over a variety of learning rates. We tested over $10$ trials and found that, with $p < 0.05$ for all cases except $\mathrm{lr} = 0.01$, the results are different.

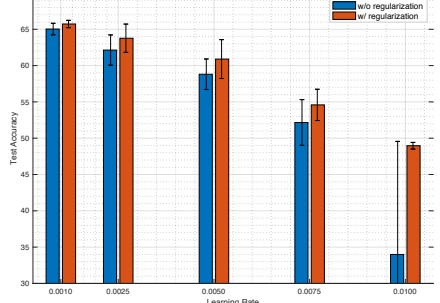

Figure 5: Effect of regularization on test accuracy for various learning rates. Our regularization is consistently able to produce higher accuracies, and this effect is more pronounced when the network has a lower test accuracy.

Our topological optimizer is able to produce the best improvements when our network is not able to converge well. These results show that our regularizer behaves as expected: the regularizer is able to recover poor training dynamics. We note that this experiment uses a simple architecture and as such, it presents a proof of concept. We do not aim for the state of the art results. Furthermore, we directly compared our approach with the generalization estimator of [CMEM20], which most closely resembles our construction. In particular, we found their method does not scale and is often numerically unreliable. For example, their methodology grows quadratically with respect to number of network weights and linearly with the dataset size, while our method does not scale much beyond memory usage with vectorized computation. Furthermore, for many of our test networks, their metric space construction (which is based off of the correlation between activation functions and used for the Vietoris-Rips complex) would be numerically brittle and result in degenerate persistent homology. These prevent [CMEM20] to be applicable in this scenario.

## 5.2 Ablation Studies

To assess the quality of our dimension estimator, we now perform ablation studies, on a synthetic data whose the ground truth ID is known. To this end, we use the synthetic experimental setting

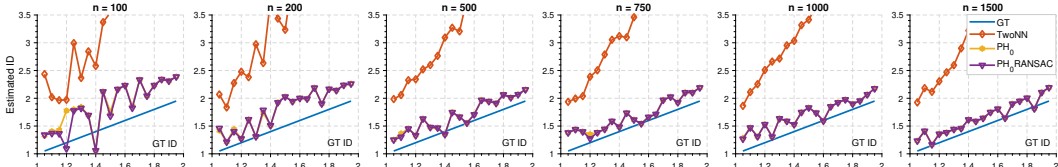

Figure 6: Estimated intrinsic dimension vs ground-truth intrinsic dimension for different dimension estimators on synthetic diffusion data. Our $\mathrm{PH}_0$ (yellow) and $\mathrm{PH}_0\mathrm{RANSAC}$ (purple) estimators coincide as the linear regression step of our computation is well behaved. We note that our persistent homology dimension estimation is able to accurately recover the ground truth.

presented in [SSDE20] (see the supplementary document for details), and we simulate a $d = 128$ dimensional stable Levy process with varying number of points $100 \leq n \leq 1500$ and tail indices $1 \leq \beta \leq 2$. Note that the tail index equals the intrinsic dimension in this case, which is an order of magnitude lower for this experiment.

**Can $\dim_{\mathrm{PH}}$ match the ground truth ID?** We first try to predict the GT intrinsic dimension running Alg. 1 on this data. We also estimate the TwoNN dimension [FdRL17] to quantify how the state of the art ID estimators correlate with GT in such heavy tailed regime. Our results are plotted in Fig. 6. Note that as $n$ increases our estimator becomes smoother and well approximates the GT up to a slight over-estimation, a repeatedly observed phenomenon [CCCR15]. TwoNN does not guarantee recovering the box-dimension. While it is found to be useful in estimating the ID of data [ALMZ19], we find it to be less desirable in a heavy-tailed regime as reflected in the plots. Our supplementary material provides further results on other, non-dynamics like synthetic dataset such as points on a sphere where TwoNN can perform better. We also include a robust line fitting variant of our approach $\mathrm{PH}_0$-RANSAC, where a random sample consensus is applied iteratively. Though, as our data is not outlier-corrupted, we do not observe a large improvement.

**Effect of $\alpha$ on dimension estimation.** While our theory requires $\alpha$ to be smaller than the intrinsic dimension of the trajectories, in all of our experiments we fix $\alpha = 1.0$. It is of curiosity whether such choice hampers our estimates. To see the effect, we vary $\alpha$ in range $[0.5, 2.5]$ and plot our estimates in Fig. 7. It is observed (blue curve) that our dimension estimate follows a U-shaped trend with increasing $\alpha$. We indicate the GT ID by a dashed red line and our estimate as a dashed green line. Ideally, these two horizontal lines should overlap. It is noticeable that, given the oracle for GT ID, it might be possible to *optimize for* an $\alpha^\star$. Yet, such information is not available for the deep networks. Nevertheless, $\alpha = 1$ seems to yield reasonable performance and we leave the estimation of a better $\alpha$ for future work. We provide additional results in our supplementary material.

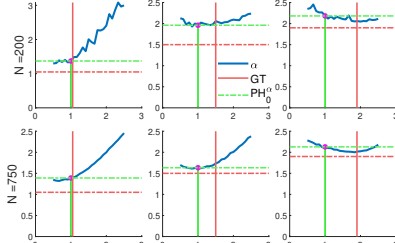

Figure 7: $\dim_{\mathrm{PH}}$ estimate versus various $\alpha$ on the synthetic diffusion data. Our estimate of $\alpha = 1$ provides a very good estimate for a wide variety of intrinsic dimensions.

## 6   Conclusion

In this paper, we developed novel connections between $\dim_{\mathrm{PH}}$ of the training trajectory and the generalization error. Using these insights, we proposed a method for estimating the $\dim_{\mathrm{PH}}$ from data and, unlike previous work [SSDE20], our approach does not presuppose any conditions on the trajectory and offers a simple algorithm. By leveraging the differentiability of PH computation, we showed that we can use $\dim_{\mathrm{PH}}$ as a regularizer during training, which improved the performance in different setups.

**Societal Impact and Limitations.** We believe that our study will not pose any negative societal or ethical consequences due to its theoretical nature. The main limitation of our study is that it solely considers the terms $E_\alpha^0$, whereas PH offers a much richer structure. Hence, as our next step, we will explore finer ways to incorporate PH in generalization performance. We will further extend our results in terms of dimensions of measures by using the techniques presented in [CDE+21].

## Acknowledgements

Umut Şimşekli's research is supported by the French government under management of Agence Nationale de la Recherche as part of the "Investissements d'avenir" program, reference ANR-19-P3IA-0001 (PRAIRIE 3IA Institute).

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
