# Intrinsic Dimension, Persistent Homology and Generalization in Neural Networks
## Supplementary Material

**Tolga Birdal**
Stanford University
tbirdal@stanford.edu

**Aaron Lou**
Stanford University
aaronlou@stanford.edu

**Leonidas Guibas**
Stanford University
guibas@cs.stanford.edu

**Umut Şimşekli**
INRIA & ENS – PSL Research University
umut.simsekli@inria.fr

## Abstract

This document supplements our main paper entitled *Intrinsic Dimension, Persistent Homology and Generalization in Neural Networks* as follows: (i) Sec. S1 firsts gives some of the formal definitions and interpretations omitted from the main paper due to space limitations. Next, it involves a discussion and contrasts our dimension estimator against the commonly used ones. Finally, it provides additional details into the regularizer we devised in the main paper; (ii) we then provide the complement the experimental evaluations given in the main paper and present additional studies on our synthetic diffusion data.

## S1 Discussions

**Connection to minimum spanning tree dimension.** In this section, we will describe another very related notion of dimension, called the *minimum spanning tree* (MST) dimension. The MST dimension coincides with the PH dimension of order 0, and provides further insights about the semantics of the PH dimension and more information about what the 'distance-matrix' illustrations topologically represent in Figure 4 in the main document, and Figures S6 and **??** in this document.

In order to define the MST dimension formally, let us introduce the required constructs. Let $W \subset \mathbb{R}^d$ be a finite set of $K$ points, $W = \{w_1, \ldots, w_K\}$. We consider a *weighted fully connected graph* $\mathcal{G}$ generated from $W$, such that the vertices of the graph are the points in $W$, i.e., $\{w_1, \ldots, w_K\}$, and the edge between the vertices $w_i$ and $w_j$ is set to the Euclidean distance between the vertices, i.e., $\|w_i - w_j\|$. Given this graph, we will consider spanning trees over $\mathcal{G}$, where a spanning tree over $\mathcal{G}$ is a tree whose nodes cover the vertices of $\mathcal{G}$ and the weights between the nodes are (still) the Euclidean distance between the nodes.

In the rest of the section, with a slight abuse, we will use the notation *"T is a spanning over the set $W$"*, meaning that $T$ is a spanning tree over the graph $\mathcal{G}$ generated by the set $W$. The notation $e \in T$ will imply that $e$ is an edge in the tree $T$.

**Definition 1** (Total Edge Length). *Let a $W \subset \mathbb{R}^d$ be a finite set and $T$ be a spanning tree over $W$. Then, the total edge length of $T$ for $(0 < \alpha < d)$ is defined as:*

$$E_\alpha(T) = \sum_{e \in T} |e|^\alpha = \sum_{(w_i, w_j) \in T} \|w_i - w_j\|^\alpha. \tag{S1}$$

**Definition 2** (Minimum Spanning Tree (MST)). *The minimum spanning tree (MST) is obtained as the minimizer* $\mathrm{MST}(W) = \arg\min_T E_1(T)$.

35th Conference on Neural Information Processing Systems (NeurIPS 2021), Sydney, Australia.

Now, we are ready to define the MST dimension.

**Definition 3** (MST Dimension [KLS06])**.** *For a bounded metric space $\mathcal{W}$, the MST dimension is defined as the infimal exponent $\alpha$ s.t. $E_\alpha(\mathrm{MST}(W))$ is uniformly bounded for **all finite point sets** $W \subset \mathcal{W}$. Formally,*

$$\dim_{\mathrm{MST}}\mathcal{W} := \inf\left\{\alpha \,:\, \exists C \in \mathbb{R}_+;\ E_\alpha(\mathrm{MST}(W)) \leq C, \quad \forall W \subset \mathcal{W},\, \mathrm{card}(W) < \infty\right\} \quad (S2)$$

*where $\mathrm{card}(W)$ denotes the cardinality of $W$.*

By the seminal work of Kozma *et al*. [KLS06] (also [Sch19]), we have that, for all bounded $\mathcal{W} \subset \mathbb{R}^d$:

$$\dim_{\mathrm{MST}}\mathcal{W} = \dim^0_{\mathrm{PH}}\mathcal{W} = \dim_{\mathrm{Box}}\mathcal{W}. \quad (S3)$$

This relation provides further intuition over what topological properties of $\mathcal{W}$ are taken into account by the MST dimension, hence the PH dimension. We observe that, given a finite set of points $W \subset \mathcal{W}$, the MST dimension summarizes the topology of $\mathcal{W}$ by the graph generated by $W$, which is essentially the distance matrix given in Figure 4 in the main paper, and Figures S6 and **??** here. Then, the growth rate of the total edge lengths of the MSTs over $W \subset \mathcal{W}$ determine the intrinsic dimension of $\mathcal{W}$. This relationship provides a more formal illustration to the visualization provided in the experiments section: the structure of the distance matrices (e.g., the clustering behavior) has indeed a crucial role in the intrinsic dimension, hence in generalization performance by [SSDE20]. Therefore, a fine inspection of the distance matrix obtained through the SGD iterations can be predictive of the generalization performance.

**On the intrinsic dimension estimators.** The notion of *intrinsic dimension* (ID) is introduced by Robert Bennett [Ben69] in the context of signal processing as the minimum number of parameters needed to generate a data description so that (i) information loss is minimized, (ii) the 'intrinsic' structure characterizing the dataset is preserved. Following its definition, different ID estimators were proposed with different design goals such as computational feasibility, robustness to high dimensionality & data scaling or accuracy [CS16]. Most of these estimators either are strictly related to Hausdorff dimension (HD) [Hau18] or try to reconstruct the underlying smooth manifold, leading to the broader notion of *manifold dimension* or *local intrinsic dimension*. On one hand, the former, HD, is very hard to estimate in practice. On the other hand, explicitly reconstructing the manifold or its geodesics as in ISOMAP [TDSL00] can be computationally prohibitive.

These challenges fostered the development of a literature populated with measures of various intrinsic aspects of data or its samples [FdRL17, CCCR15, Sch09]. Geometric approaches like *fractal* measures usually replace the Hausdorff dimension with a lower or upper bound aiming for scalable algorithms with reasonable complexity. Kegl *et al*. [Kég02] try to directly estimate an upper bound, the Box Counting or Minkowski dimension. Though, their Packing Numbers are not multiscale and have quadratic complexity in the cardinality of the data. Correlation dimension (*strange attractor dimension* in chaotic dynamical systems) on the other hand is a lower bound on the Box Counting dimension [GP04] and is efficient to compute. Another fractal dimension estimator, Maximum Likelihood Estimation (MLE) dimension [LB05] estimates the expectation value of Correlation dimension. Finally, Information dimension [Ish93] measures the fractal dimension of a probability distribution. As we show in Thm. 1 of the main paper, our PH dimension measures the upper bound, Minkowski Dimension exactly and is related to the generalization in neural networks.

The other line of work trying to estimate a *manifold dimension* makes the assumption that the dataset or in our case the weights of neural networks lie on a smooth and compact manifold. The *geodesic minimum spanning tree length* of Costa and Hero try to approximate the geodesics of this supposedly continuous manifold [CH03]. Lin and Zha [LZ08] reconstruct the underlying *Riemannian Manifold* with its local charts through 1D simplices ($k$-nearest edges). The dimension is then estimated by performing a local PCA [Jol86]. It is interesting and noteworthy that such parallel track of works based upon a discretization of the underlying smooth manifold can yield algorithms very similar to ours. Though, we are not aware of well established connections between these fractal based and manifold based approaches. Evaluating the performance of these manifold dimension methods is out of the scope of this paper since unlike our method, they are not explicitly linked to generalization.

We are, however, curious to see how the estimators related to Box Counting dimension perform in our setting. To do that, we use a similar synthetic diffusion process as we did in the main paper. This allows us to have access to the ground truth dimension as the tail index of a $\beta$-stable process (see

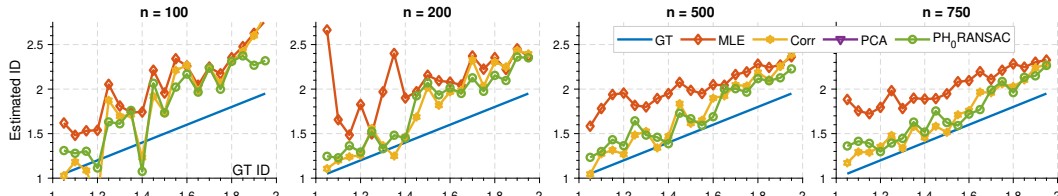

Figure S1: Comparing different dimensions estimators for different number of sample points $n$. We simulate the point cloud data as the trajectories of a diffusion using $\beta$-stable processes in $d = 128$ ambient dimensions. The tail index $\beta$ mapped to the $x$-axis of all plots corresponds to the ground truth (GT) intrinsic dimension. $\beta \to 1$ models a heavy-tailed process whereas $\beta \to 2$ is a Brownian motion. Our method and correlation dimension can capture the intrinsic properties of this data. Yet, our method performs slightly better as it is a theoretically grounded measure of the tail index.

below for the definition). In Fig. S1 we put correlation dimension (Corr), MLE dimension (MLE) and our dimension ($\mathrm{PH}_0\mathrm{RANSAC}$) at test. For sanity check, we also include an *eigen-value* based ID estimator *PCA Dimension* [Jol86]. It is seen that all of these methods over-estimate the dimension in small-data regime and approach the correct dimension as $n$ grows. As expected, our method – explicitly connected to the tail index – outperforms the other methods. Correlation dimension performs very reasonably in this experiment, only slightly worse. It is also noteworthy that the PCA Dimension estimates are not well correlated with GT tail index (they do not even fit the axis limits of our plot) and therefore cannot be used in our framework to measure generalization in deep networks. This is because (i) they are expected to fail on nonlinear manifolds such as diffusion or network trajectories; (ii) they do not measure per se the fractal dimension, which are useful in evaluating the model order in time series data or nonlinear dynamical systems.

Among different dimension estimators, the Two-NN estimator [FdRL17] has recently been found to be of practical value in the context of measuring the intrinsic dimension of data representations, or the layers of the network [ALMZ19]. However, this estimator is based upon the assumption of approximately constant density on the length-scale defined by the typical distance to the second neighbor. In the main paper we showed that this assumption breaks for the processes that exhibit heavy tails, such as the trajectories of deep network training algorithms, making Two-NN a bad estimator. In Fig. S2, we show that for data where this assumption holds – such as randomly sampled points on a $d$-dimensional hypersphere, Two-NN can be a good estimator. Yet, our estimator works both in this case and in the case of heavy tailed diffusion-like processes.

**On the interpretation of Figure 4 – middle panel.** We shall note two important points regarding the plots given in Figure 4, middle panel:

1. The slope of the fitted line essentially determines the intrinsic dimension (see the paragraph after Proposition 2 to see the mathematical relation) and we illustrate this behavior visually in the plots: the slope of the data computed by using persistent homology determines the intrinsic dimension and hence determines the generalization gap; which is arguably a surprising result.

2. Even though we fit a line to the empirical data, it does not automatically mean that the empirical data should have a clear linear trend. These plots further demonstrate that we fit a line to a data, which has a very strong linear trend, hence our estimations are not jeopardized by noise or model mismatch.

**On the interpretation of distance matrices.** The columns and rows of the distance matrices shown here and in the main paper are organized with respect to the iteration indices after convergence: we first train the networks until convergence, and then run an additional 200 iterations near the local minimum. Then, the $(i, j)$-th entry of a distance matrix corresponds to the Euclidean distance between the $i$-th iterate and the $j$-th iterate (of these additional 200 iterations).

Though we do not yet have a rigorous proof, we believe that the qualitative difference in these diagrams is due to the heavy-tailed behavior of the SGD algorithm [SSG19, GSZ21, HM20]. Let us illustrate this point with a simpler example: consider the Levy $\beta$-stable process used in Section 5.2 of the main paper. This is a well-known heavy-tailed process which becomes heavier-tailed when

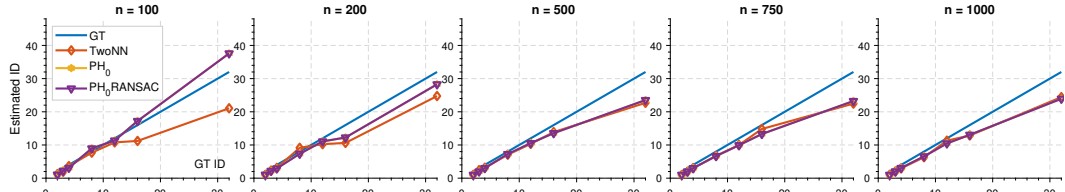

Figure S2: Results of different dimension estimators for synthetic sphere data. GT, TwoNN, $PH_0$ and $PH_0RANSAC$ denote the ground truth, TwoNN estimator [FdRL17], our standard persistent homology estimator and its robust line fitting variant, respectively.

the parameter $\beta$ decreases. A classical result in probability theory [BG60] shows that the Hausdorff dimension of the trajectories of this process is (almost surely) equal to its tail-index $\beta$ for any $d \geq 2$. This means that as the process gets heavier-tailed, its intrinsic dimension decreases.

On the other hand, if we investigate the geometric properties of heavy-tailed processes, we mainly observe the following behavior: the process behaves like a diffusion for some amount of time, then it makes a big jump, then it repeats this procedure. For a visual illustration, we can recommend Figure 1 in [SSDE20]. Finally, this *diffuse+jump* structure creates clusters, where each cluster is created during the *diffuse* period and a new cluster is initiated at every large *jump*.

Now, coming back to the question of interpreting the distance matrices, the non-uniform pixelations in the distance matrices are well-known indications of clusters in data: in the top row of Figure 4, the large dark squares indicate different clusters, and we observe that these clusters become more prominent for smaller PH dimensions. To sum up:

(i)     SGD can show heavy-tailed behavior when the learning-rate/batch-size is chosen appropriately [SSDE20, GSZ21].

(ii)    Heavy-tails result in a topology with smaller dimension [SSDE20], and creates a clustering behavior in the trajectories.

(iii)   The clustering behavior results in non-uniform pixelations in the distance matrices and gaps in the persistence diagrams.

## S2    Details of Experimental Evaluation

**Analysis**    For our analysis, we train with the following architectures. For basic networks, we include fully connected models with 5 (fcn-5) and 7 (fcn-7) layers and a 9-layer convolutional network (cnn-9). All networks have ReLu activation. We also include several more standard networks such as AlexNet [KSH12]. We train using variants of SGD without momentum or weight-decay. We trained the networks with different step-sizes in the range $[0.001, 0.1]$ and batch-sizes in the set $\{64, 100, 128\}$. We trained all models until convergence. We implemented our code in PyTorch and conducted the experiments on 4 GeForce GTX 1080 GPUs. We used the classical CIFAR10 and MNIST datasets and for measuring the training and test accuracies, we use the standard training-test splits.

**The $\beta$-stable Levy process**    In our experiments, we illustrated our approach on estimating the intrinsic dimension of $\beta$-stable Levy processes[1], whose definition is given as follows.

For $\beta \in (0, 2]$, a $\beta$-stable Lévy process $\{L_t^\beta\}_{t \geq 0}$ in $\mathbb{R}^d$ with the initial point $L_0^\beta = 0$, is defined by the following properties:

(i)     For $N \in \mathbb{N}$ and $t_0 < t_1 < \cdots < t_N$, the increments $(L_{t_i}^\beta - L_{t_{i-1}}^\beta)$ are independent for all $i$.

(ii)    For any $t > s > 0$, $(L_t^\beta - L_s^\beta)$ and $L_{t-s}^\beta$ have the same distribution, and $L_1^\beta$ has the characteristic function $\exp(-\|\omega\|^\beta)$.

(iii)   $L_t^\beta$ is continuous in probability, i.e., for all $\delta > 0$ and $s \geq 0$, $\mathbb{P}(|L_t^\beta - L_s^\beta| > \delta) \to 0$ as $t \to s$.

---

[1]This processes are often called the $\alpha$-stable processes, where $\alpha$ denotes the stability index of the process [ST94]. However, to avoid confusion with the parameter $\alpha$ in $E_\alpha$, we denote the stability index with the symbol $\beta$.

**Algorithm 1:** Topological Regularization Training

**1 input** : The set of data $\mathcal{D} = \{(x_i, y_i)\}_{i=1}^{D}$, gradient update method gd, number of networks for dimension computation $K$, regularization constant $\lambda$, number of steps $N$, batch size $b$, regular training loss $\mathcal{L}$

**2 output** : Regularized trained network $W$

**3** $n \leftarrow 0$, initial network $W_0$, network list $Q$. **while** $n \leq N$ **do**

**4** | $\{x_i, y_i\}_{\mathcal{S} \subset [D]} \leftarrow \text{sample}(W, b)$ `// random sampling`

**5** | $Q \leftarrow Q \cup \{W_n\}$ `// append the last network`

**6** | $\mathcal{L}_{top} \leftarrow \mathcal{L}(x_i, y_i) + \lambda \cdot \text{compute\_dims}(Q, K)$ `// Compute loss of last K models`

**7** | $W_{n+1} = \text{gd}(W_n, \nabla_{W_n} \mathcal{L}_{top}), n \leftarrow n + 1$ `// update weights w.r.t. gradients`

**8** Return $W_N$

---

The stability index (also called the tail-index) $\beta$ determines the tail-behavior of the process (e.g., the process is heavy-tailed whenever $\beta < 2$) [ST94]. On the other hand, interestingly $\beta$ further determines the geometry of the trajectories of the process, in the sense that, [BG60] showed that the Hausdorff dimension of the trajectory $\{\mathrm{L}_t^{\beta}\}_{t \in [0,T]}$ is equal to the tail-index $\beta$ almost surely. Similar results have also been shown for the box dimension of the trajectories as well [Fal04]. Thanks to their flexibility, $\beta$-stable processes have been used in the recent deep learning theory literature [SSG19, ZFM+20, SSDE20].

## S3 Additional Evaluations

**Use of different robust functions.** Our dimension estimate is essentially the slope of a line fit to the log-sequence of total edge lengths. Since a slight change in the slope can directly influence the resulting dimension, it is of interest to see the effect of different line estimators. To this end, we evaluate the naive least squares (LS) fit as well as a series of methods that are robust to the outliers. In particular, we deploy random sample consensus (RANSAC) [FB81] as well as Huber [Hub92], Tukey [BT74] robust M-estimators. As seen in Fig. S3, the difference

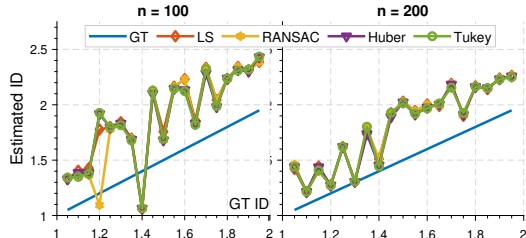

Figure S3: Effect of different robust functions on dimension estimation.

between them in dimension estimation is negligible. This indicates that, as expected, the data points composed of total edge lengths are not corrupted by outliers – every iterate (network) in the trajectory yields a valid data point. We used the RANSAC estimator as it gave a marginally better estimate.

**Effect of regularization on the intrinsic dimension.** For our regularization procedure, we train the LeNet [LBBH98] architecture on Cifar10 with a batch size of 128. We again optimize with a constant step size SGD varying the learning rate from 0.01 to 0.1. The exact training procedure is summarized in Alg. 1 where we show how to apply our dimension constraint in a sliding fashion.

Here, we test to see if our regularization would be able to control the intrinsic dimension of the training trajectories. In particular, we calculate the intrinsic dimension of our regularization experiments and report them in Fig. S4. It is noticeable that our topological regularizer is indeed able to decrease the intrinsic dimension for different training regimes with different learning rates. This property is well reflected to the advantage in Fig. 5 of the main paper.

| Learning rate | 0.001 | 0.0025 | 0.005 | 0.0075 | 0.01 |
|---|---|---|---|---|---|
| Unregularized | 3.21 | 3.35 | 3.54 | 3.67 | 4.2 |
| Regularized | 3.13 | 3.25 | 3.30 | 3.43 | 3.75 |

Figure S4: The intrinsic dimensions (ID) of unregularized vs topologically regularized network training. Across the different learning rates, our regularized network has a better behaved (lower) ID.

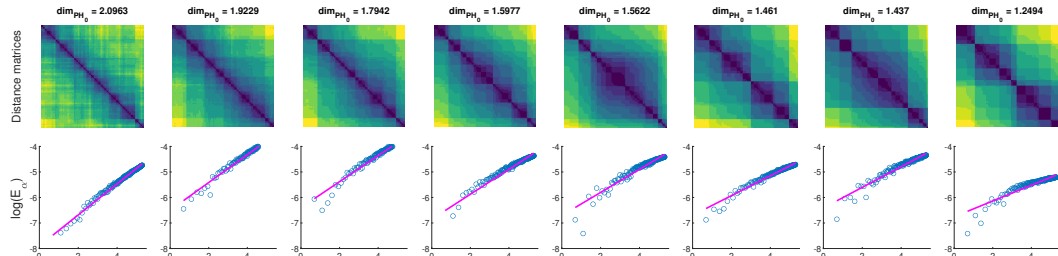

Figure S5: Visualization for cnn-9 network on MNIST dataset sorted by persistent homology dimension. (**top**) the distance matrices computed between the network weights corresponding to the last 200 iterates. (**bottom**): the behavior of the logarithmic $\alpha$-weighted lifetime sums $\log E_\alpha^0$, derived from the persistent homology with respect to $\log(n)$.

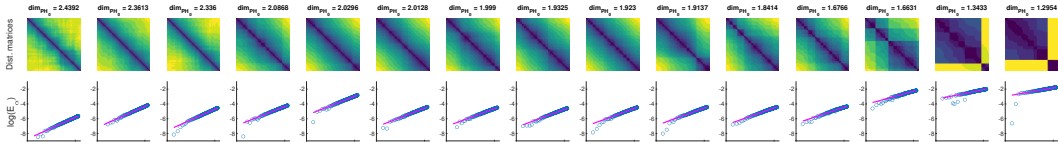

Figure S6: Visualization for VGG-like cnn-9 network on Cifar10 dataset sorted by PH dimension. Please see Fig. S5 for more information on the plots.

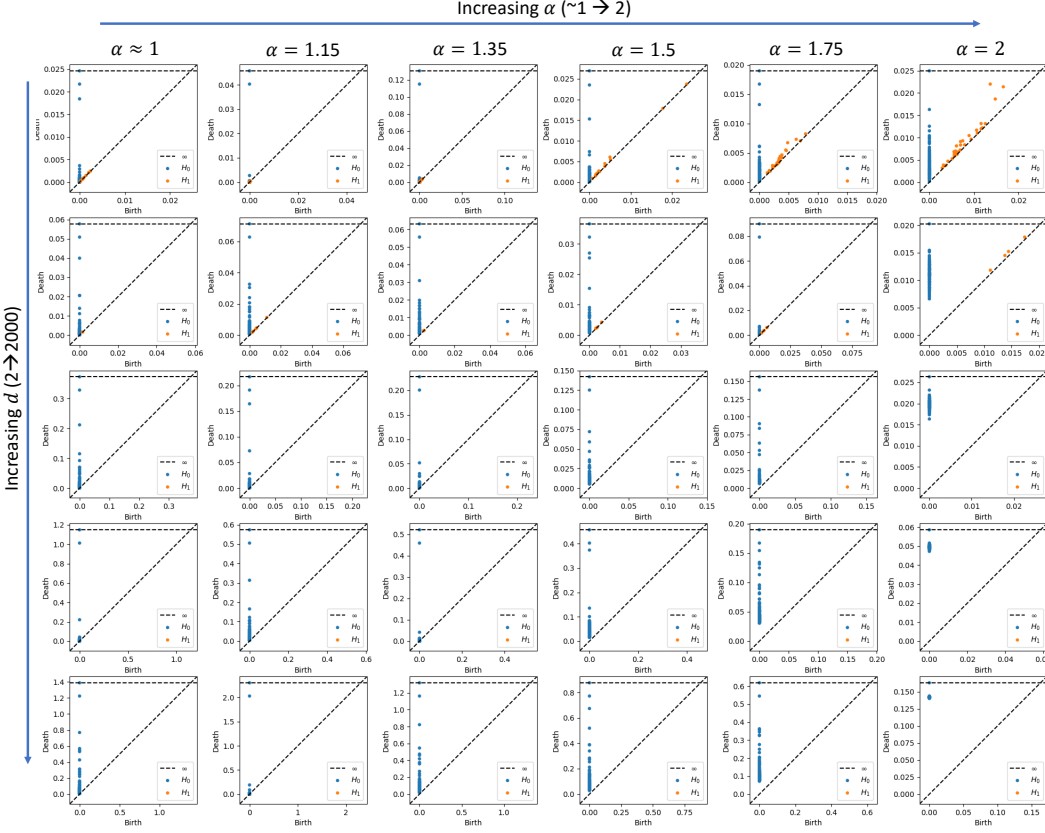

Figure S7: Persistence Diagrams for the diffusion processes of varying ambient dimension and intrinsic dimension (tail index).

**Further visualizations on cnn-9 network.** In the main paper we have visualized the distance matrices, total edge lengths $E_\alpha^0$, and the persistence diagrams for the 7-fcn network. We have observed a clustering effect with decreasing persistent homology dimension. One could question

whether the same effect is observable in other networks and for other datasets. Hence, we now visualize the same quantities for our cnn-9 (VGG-like) network both on Cifar10 and MNIST datasets. Figures S5 and S6 plot the distance matrices and $\log E_\alpha^0$-values in a fashion identical to the main paper for MNIST and Cifar10 datasets, respectively. Note that in both figures there is a clear *pixelation* effect on the distance matrices as the PH-dimension decreases. This indicates a clustering as expected from a diffusion process. We measures this by the slope of the line as shown in the bottom rows. We conclude that the behavior reported in the main paper is consistent across datasets and networks.

**Further visualizations on synthetic data.** Last but not least, we show the effect of the change in intrinsic dimension to the persistence diagram. To this end, we synthesize point clouds of different ambient and intrinsic dimensions from the aforementioned diffusion process. We then compute the persistence diagram of all the point clouds and arrange them into a grid as shown in Fig. S7. This reveals two observations: (i) As $\alpha \to 1$, the persistence diagram gets closer (in the relative sense of the word) to the diagonal line while forming distinct clusters of points. Note that as death is always larger than birth, by definition, these plots can only be upper-triangular. As we approach $\alpha \to 2$, the process starts resembling a Brownian motion and we observe a rather dispersed distribution while the points move away from the diagonal line (relatively), at least for the case we are interested in, $H_0$.