# OpenReview forum: "Intrinsic Dimension, Persistent Homology and Generalization in Neural Networks"
_NeurIPS.cc/2021/Conference — NeurIPS 2021 Poster_

### Official Review · Reviewer_J8S7 · 2021-07-08

**Rating:** 2
**Confidence:** 4

**Summary:**

The paper proposes to use an estimator of the intrinsic dimension (ID)  based on topological data analysis for two goals, both important in neural network theory: (1) as a proxy of  the test accuracy, which allows estimating it without performing explicitly any validation and (2) as a regularizer, by adding an ID-dependent term to the loss. A part of the paper is devoted to derive (or recall from the literature) rigorous properties of this ID estimator.


**Limitations And Societal Impact:**

the limitations of the approach are discussed only shortly, possibly due to length constraints. But I do not see this as the main  problem of the poper.

**Main Review:**

The quality of the presentation is in general poor and the results of the numerical tests seem to me already present in the literature, and  in a case possibly flawed.

Fig 2: the x axis are not labeled. Fig 3: the dataset on which the tests are performed is not specified. If the x axis (not labeled) is  the difference between test and train accuracy, then this  ranges between 30 and 40. IThese differences are enormous, for any dataset mentioned in the paper, pointing to possible flaws in the model. Fig.  4 is unreadable, even by magnifying it. Its message is, at least to me, obscure. In the middle panel we see points approximately lying on a line, which is floating up and down across the panels. What do we learn from this? the bottom lines the two sets of points are labeled H_0 and infinity. What does this mean? Fig 7 is also unreadable. The axis labels are missing. The different panels are also not labeled.

The idea of regularizing learning by controlling the ID is not new. It was introduced in ref [MA] https://arxiv.org/abs/1806.02612  (2018). This is a key reference that should be cited and discussed. The results illustrated in the manuscript on this important point are not really convincing: in fig 5 it is shown that with the learning rate which allows obtaining the best test accuracy the effect of ID regularization  is practically zero (65 % with and without regularization). Moreover, an accuracy of 65 % in cifar10 is way below the state of the art for this dataset which is 95 % for convolutional NNs, and almost 99 % for architectures exploiting transformers (see for example https://paperswithcode.com/sota/image-classification-on-cifar-10). Also this points to possible flaws.

The intrinsic dimensions reported in fig 2 and 3 are of order 2, while those reported in ref ALMZ19 and [MA] range between 10 and 100, for the Imagenet dataset. The reason for this qualitative discrepancy should be discussed.


**Time Spent Reviewing:**

4

---

> ### Author Response · Authors · 2021-08-10
> **Author Response to Reviewer J8S7**
>
> We thank the reviewer for their time and efforts invested in our paper. However, we regret to mention that the review seems to be affected by a fundamental misunderstanding which has led to a surprisingly harsh judgement that largely contradicts all of the other reviews.
>
> We will clarify the points that might have caused the misunderstandings and address all the remaining criticisms, which, we must underline, are only minor (e.g., relabeling the figures), can be very quickly implemented in a revision, and could/should not be used for justifying a “strong rejection” decision.
>
> Under the light of all these explanations, we hope that the reviewer could reconsider their overall score.
>
> **Novelty:**
>
> *The idea of regularizing learning by controlling the ID is not new. It was introduced in ref [MA] https://arxiv.org/abs/1806.02612 (2018). This is a key reference that should be cited and discussed.*
>
> We suspect that the reviewer had an important misconception here, which seems to have affected the overall review.
>
> To clarify this point, we shall start by emphasizing that “intrinsic dimension” is merely a “concept”, and is not a well-defined mathematical object at all. In fact, there is a zoo of different mathematical definitions for intrinsic dimension, which often do not coincide. For instance, we have
>
> * Hausdorff dimension
> * (Upper) Box dimension
> * Correlation dimension
> * Information dimension
> * Algebraic dimension
>
> to name just a few. All these mathematical objects have very different definitions and semantics, and more importantly **their value can be vastly different from each other**. For instance the Hausdorff dimension of $R^d$ is just $d$; however, the upper box dimension of $R^d$ is $\infty$.
>
> Hence, the sentence “regularizing learning by controlling **the ID**” is rather misleading, since there is no single definition of ID. Each notion of ID has a different semantics and treating each notion of ID requires different mathematical/computational tools.
>
> The paper by Ma et al. is definitely relevant and we will cite/discuss it in the revised version. We thank the reviewer for this pointer. However, like all the other studies using “intrinsic dimension” cited in Section 2, Ma et al. as well use a (yet another) notion of intrinsic dimension, **which has no rigorous connections to the generalization error**. In this sense, we argue that the existing techniques in the literature are rather heuristics, and our main contribution is to bridge this gap with very little theoretical assumptions on the problem.
>
> On another note, what we propose in the paper is controlling the intrinsic dimension of **training trajectories** (rather than a single iterate), which, to our knowledge, is developed for the first time in our work.
>
> To sum up: our contribution is **not** to control an arbitrary ID during training (which wouldn’t be novel, and probably is the main confusion of the reviewer); it is to **identify** a notion of ID **which can be rigorously linked to generalization gap**, and to develop **mathematically grounded computational tools** for computing its value and incorporating it to the training process in a novel way. We hope that this explanation clarifies the confusion, we would be happy to clarify any further concerns/questions in the discussion period.
>
> Furthermore, we suspect that the same confusion has yielded the following comment:
>
> *The intrinsic dimensions reported in fig 2 and 3 are of order 2, while those reported in ref ALMZ19 and [MA] range between 10 and 100, for the Imagenet dataset. The reason for this qualitative discrepancy should be discussed.*
>
> As we described above, the notion of ID that was used in [ALMZ19] and Ma et al. were completely different mathematical objects compared to our notion, which is the PH dimension. Hence, as we exemplified above, it is very natural that they might attain different values.
>
> Nevertheless, we shall reiterate that, as opposed to our approach, there is no connection between the ID notions presented in [ALMZ19] and Ma et al., and the generalization error. As we explained in detail in the paper, our approach is an extension of [SSDE20], and our ID estimates are in agreement with their results. Yet, our framework does not require several of their theoretical assumptions and looks at the problem from a novel, topological data analysis point of view.
>
> We will expand and clarify our discussion on this matter in the revision.
>
>
> **Clarity:**
>
> *The quality of the presentation is in general poor.*
>
> We are sorry and rather surprised to see that the reviewer found the presentation of the paper poor. Since our paper combines tools from three distinct mathematical fields, namely, statistical learning theory, fractal geometry, and topological data analysis, we had to make a significant effort (also acknowledged by some other reviewers) to make our notation thorough and consistent, and our paper accessible to a wider audience.
>
> We believe that the criticisms regarding clarity are quite minor. As acknowledged by *Reviewer pbGK*, such issues can be fixed easily in the final version. Our detailed responses are given as follows.
>
> *Fig 2: the x axis are not labeled.*
>
> We agree with the reviewer. However, the caption of Figure 2 starts with “PH-dimension vs test accuracy for different models and datasets”, and the x-axis of the top left graph is labelled. We chose to design it this way to reduce clutter, but, if the reviewer prefers, we will update the final figures with this information on all axes.
>
> The same argument applies to Figure 7, except that there is a typo in the legend: $\alpha$ should be replaced by $E_\alpha$.
>
> *Fig 3: the dataset on which the tests are performed is not specified.*
>
> We disagree with this point, as the dataset information is provided on the labels of y-axes of the figures. In particular, the top row is Cifar10 and the bottom row is Cifar100. However, we understand that our current labels may be slightly hard to distinguish, and will update the caption to provide this information more clearly.
>
> *If the x axis (not labeled) is the difference between test and train accuracy, then this ranges between 30 and 40. IThese differences are enormous, for any dataset mentioned in the paper, pointing to possible flaws in the model.*
>
> We agree that the differences are quite large, although we stress that this is an intentional result of our training regiment. In particular, we analyze the “bare bones” dynamics of several canonical optimizers without additional regularization (which would produce a lower test gap). This is in line with previous work [1] and allows us to isolate and control the intrinsic dimension (as a function of learning rate only) for comparison with the test error.
>
>
> *Fig. 4 is unreadable, even by magnifying it. Its message is, at least to me, obscure. In the middle panel we see points approximately lying on a line, which is floating up and down across the panels.*
>
> We understand the reviewer’s concern and we will make the font larger for a better reading experience. Regarding the meaning of the middle panel: there are two take-home messages of these plots:
>
> * 1- The **slope** of the fitted line essentially determines the intrinsic dimension (see the paragraph L225 to see the mathematical relation) and we illustrate this behavior visually in the plots: the slope of the data computed by using persistent homology determines the intrinsic dimension and hence determines the generalization gap; which, to us is quite an unexpected result.
>
> * 2- Even though we fit a line to the empirical data, it does not automatically mean that the empirical data has a linear trend. These plots further demonstrate that we fit a line to a data, which has a very strong linear trend, hence our estimations are not jeopardized by noise of any sort.
>
> We will emphasize these points more clearly in our revision to avoid such a confusion.
>
> *the bottom lines the two sets of points are labeled H_0 and infinity. What does this mean?*
>
> This visualization is very standard in the persistent homology literature and is the default way RIPSER visualizes persistence diagrams. Hence, we omitted a detailed discussion on persistence diagrams, mainly due to space constraints.
>
> In a nutshell, these diagrams are used for summarizing the geometry of the space in a qualitative manner. The infinity line indicates the diagonal. As births are always less than deaths, data points can never fall below that line. The positioning of the points with respect to this line are descriptive. For instance, the farther they are, the larger the holes. We will include a detailed description of persistence diagrams in the supplementary document.
>
>
> **Experimental results:**
>
> We will respond to the following two comments simultaneously, as they concern a similar point.
>
> *The difference between test and train accuracy, then this ranges between 30 and 40. IThese differences are enormous, for any dataset mentioned in the paper, pointing to possible flaws in the model.*
>
> *The results illustrated in the manuscript on this important point are not really convincing: in fig 5 it is shown that with the learning rate which allows obtaining the best test accuracy the effect of ID regularization is practically zero (65 % with and without regularization). Moreover, an accuracy of 65 % in cifar10 is way below the state of the art for this dataset which is 95 % for convolutional NNs, and almost 99 % for architectures exploiting transformers*
>
> Let us start by noting that the purpose of our experiments is to not attain state of the art accuracy; our goal is rather to **validate** our theoretical and computational contributions.
>
> The main reason why we obtained a large training-testing gap in these experiments is not due to a flaw, we invite the reviewer to verify it with the code provided in our submission (which is not complicated).

---

> > ### Author Response · Authors · 2021-08-10
> > **Author Response to Reviewer J8S7 Cont'd**
> >
> > The cause of the gap is rather due to our choice of network architecture: our chosen network of LeNet-5 does not trivialize Cifar10 and, as such, controlling the intrinsic dimension can produce visible changes in the test error, which is more preferable for illustration. Since our goal is to **illustrate that the estimated ID is informative of the generalization gap**, and illustrate that regularizing the ID of the **trajectories** can be beneficial as a proof of concept, we believe that our experiments adequately serve this purpose.
> >
> > We will include the motivation for network choice in our paper.
> >
> > As the review mentions, a learning rate of 0.001 does in fact produce a less significant change in test error. In concert with our theory, we indeed expect this; this learning rate results in a small intrinsic dimension which results in a lower test gap. Since intrinsic dimension is bounded below by one, we hypothesized that optimizing it here would produce diminishing returns (which our results show). We will rewrite our analysis of this experiment to be more clear about our intent and how this relates to our previously developed theory.
> >
> > We also invite the reviewer to examine our newer statistics for Figure 5 (which are in the response to reviewer MbMP), as they may help elucidate the significance of our method. In particular, we note that even for the accuracy 65% case, our optimizer does in fact produce a moderately significant improvement.

---

### Official Review · Reviewer_PP68 · 2021-07-10

**Rating:** 7
**Confidence:** 4

**Summary:**

The relationship between generalized loss of NN and Intrinsic Dimension by Persistent homology is mathematically proved. They also propose a normalization term based on the theory presented. In addition, we experimentally verify the theoretical results and confirm the effectiveness of the proposed algorithm.

**Limitations And Societal Impact:**

I don't believe there are any negative effects, and I think this is something that can be developed in the future, but for practical purposes, it needs more testing.

**Main Review:**

For the normalization term of NN loss, many methods exist, but many are experimental and few have been mathematically proven. Mathematical analysis is necessary for the future development of technology. This paper logically shows that the generalization of NN can be adjusted by intrinsic dimension based on persistent homology. Most of the theories presented are based on previous results, but they are well developed for the generalization of NNs. Although the proposed normalization term is a simple one, it seems to be significant just because it shows the mathematical implications. Many normalization terms have been proposed in the past. In this paper, only the effect for one problem is shown, but in practical terms, comparison with other normalization terms may be necessary. In addition, since persistent homology is generally computationally expensive, future evaluation in terms of computational complexity will be necessary.

**Time Spent Reviewing:**

8 hours

---

> ### Author Response · Authors · 2021-08-10
> **Author Response to Reviewer PP68**
>
> We thank the reviewer for their time and effort and their positive feedback. Our detailed responses are given below.
>
> *Many normalization terms have been proposed in the past. In this paper, only the effect for one problem is shown, but in practical terms, comparison with other normalization terms may be necessary.*
>
> We agree with the reviewer that there are multiple ways to measure generalization and some of the recent results can indeed be used to regularize for generalizing networks. Our regularization experiment is, for the moment, a proof of concept. Nevertheless, in the revision, we will be including some other ‘normalization terms’ such as the one of Gabrielsson et al. PMLR 2020. We thank the reviewer for bringing this up.
>
> *Since persistent homology is generally computationally expensive, future evaluation in terms of computational complexity will be necessary.*
>
> We refer the reviewer to our generalized response for comments on computational complexity. We plan on including a note on computational complexity in the final version if our paper is accepted with our additional page.

---

> > ### Comment · Reviewer_PP68 · 2021-08-17
> > **Thank you for your response**
> >
> > Sorry, I was writing my comment in the wrong place ... I deleted the previous one and rewrite it.
> >
> > Thank you for your response. I now have a clearer idea of your revision policy and future plans. Good luck revising your draft.

---

> > > ### Author Response · Authors · 2021-08-17
> > > **Thank you for your feedback**
> > >
> > > We appreciate that Reviewer PP68 finds our responses useful. We would like to emphasize that we are committed to applying the changes (which are not substantial) in the revision.

---

### Official Review · Reviewer_MbMP · 2021-07-16

**Rating:** 7
**Confidence:** 3

**Summary:**

This paper proposes a measure of generalization based on persistence homology, a standard technique of topological data analysis.

The proposed measure, the persistence homology dimension (PHD), is computed on optimization trajectories during training.

A theoretical connect between PHD and generalization is made using existing results in the literature: first the equivalence of PHD and the box-dimension [KLS06, Sch19], and second that the generalization error may be bounded under the box dimension (under the assumption that the optimization dynamics follow a Feller process).

From these main result of the paper, Proposition 1, then follows under assumption (H1) .
The authors then proceed to remove the Feller requirement in Proposition. 2 using a recent result in the mathematics literature [Sch20].

Based on these results, an algorithm for computing PHD is proposed.

The authors proceed to perform an empirical evaluation of how well PHD predicts generalization for select deep network architectures (alexnet, cnn-9, fcn-5, fcn-7), datasets (mnist, cifar-10, cifar-100), and training hyperparameters (learning rate, batch size).

The authors visualize these results in Figures 2 and 3, where in Figure 3 the authors claim that PHD "directly" correlates with generalization error across experimental settings. Linear trends are evident in the plots, however no measures of goodness of fit are given.

In addition, a novel loss regularizer is proposed based on their PHD measure. The effect on generalization of his regularizer is studied, but small differences are shown.

Ablation studies are additionally carried out.

**Limitations And Societal Impact:**

A brief but adequate discussion of societal impacts is given in Section 6.

A brief discussion of limitations is given in Section 6, which could be expanded.

**Main Review:**

This work addresses an important problem, predicting generalization, and approach it from an interesting perspective, that of TDA.

I have some doubts about the empirical evaluation of PHD's predictive ability of generalization. No quantitative measures are given. There indeed appears to be trends in Figures 2 and 3, but their strength is unclear.

Even so, as some have pointed out in the literature, correlative measures may be problematic and not indicative of a causal relationship [0,1].

For instance in [0], the authors write:
*Correlation with Generalization: Evaluating measures based on correlation with generalization is very useful but it can also provide a misleading picture. To check the correlation, we should vary architectures and optimization algorithms to produce a set of models. If the set is generated in an artificial way and is not representative of the typical setting, the conclusions might be deceiving and might not generalize to typical cases... Another pitfall is drawing conclusion from changing one or two hyper-parameters (e.g changing the width or batch-size and checking if a measure would correlate with generalization). In these cases, the hyper-parameter could be the true cause of both change in the measure and change in the generalization, but the measure itself has no causal relationship with generalization. Therefore,one needs to be very careful with experimental design to avoid unwanted correlations.*

Can the authors justify their choice of empirical methodology, over the Kendall's rank correlation and conditional mutual information as suggested by [1]?

Using the measure for regularization is a natural and interesting idea. However it's empirical evaluation appears weak in Figure 5, i.e. < +2%, except for the high learning rate case. I would like to see error bars on this experiment to gauge it's significance.

Minor notes: Typo on line 256 "netowork"

[0] Fantastic  Generalization  Measures and  Where  to  Find  Them - Jiang et al. (2019)
https://arxiv.org/abs/1912.02178

[1] NeurIPS 2020 Competition:Predicting Generalization in Deep Learning - Jiang et al. (2020)
https://arxiv.org/abs/2012.07976v1

**Time Spent Reviewing:**

4

---

> ### Author Response · Authors · 2021-08-10
> **Author Response to Reviewer MbMP**
>
> We thank the reviewer for their constructive and interesting comments. We begin by addressing the related three remarks:
>
> *The authors visualize these results in Figures 2 and 3, where in Figure 3 the authors claim that PHD "directly" correlates with generalization error across experimental settings. Linear trends are evident in the plots, however no measures of goodness of fit are given.*
>
> *I have some doubts about the empirical evaluation of PHD's predictive ability of generalization. No quantitative measures are given. There indeed appears to be trends in Figures 2 and 3, but their strength is unclear.*
>
> *Can the authors justify their choice of empirical methodology, over the Kendall's rank correlation and conditional mutual information as suggested by [1]?*
>
> It is important to understand that our results are not mere observations or correlations but are backed by key theoretical results that, arguably for the first time, connect generalization to one topological characteristic of a neural network: intrinsic dimension of training trajectories. As expected, we observe this theoretical ground in our experimental evaluation. In line with the references suggested by the reviewer [0,1] (which we now cite) we do vary architectures and optimization algorithms to generate a set of models representative of the typical settings. We also do not re-invent an evaluation methodology but consult the recipes given in the literature [SSDE20].
>
> We certainly agree with the reviewer that a single metric measuring the quality of the complexity measure is a good idea. For this reason we conducted two additional experiments, both of which are in line with Figure 3. First, for the Cifar100 dataset, AlexNet, trained with SGD achieves a mean p-value of $0.0157$ over different batch sizes. This is reasonably lower than $0.05$ indicating statistical significance. We will be providing a detailed table of these in the final publication.
>
> Next, we also evaluated the Kendall correlation coefficient (KCC) [1] as suggested by the reviewer. Similar to the p-value experiment explained above, we compute KCC for AlexNet+SGD for different batch sizes (64, 100, 128). We attain $0.933$, $0.357$, $0.733$, respectively. Note that having this positive correlation means that the test gap closes as PH-dim decreases.
>
> *Using the measure for regularization is a natural and interesting idea. However it's empirical evaluation appears weak in Figure 5, i.e. < +2%, except for the high learning rate case. I would like to see error bars on this experiment to gauge it's significance.*
>
> We understand the reviewer’s concerns and have taken action to increase our experimental evaluation. We will supplement our paper with error bars reporting the standard error. We also increased the experimental evaluation for Figure 5 with more trials. Our results are now taken over 10 trials, and we observed the same pattern from the paper for these new cases. We calculated the t-values to range from 2.7 to 1.67, and this decreases monotonically as one decreases the learning rate from 0.01 to 0.001 as expected. We note that these results range from highly significant (p-value < 0.05) to moderately significant (0.05 < p-value < 0.1).

---

> > ### Comment · Reviewer_MbMP · 2021-08-16
> > **Reviewer MbMP Response to Author**
> >
> > I found the author's response regarding my questions about empirical methodology convincing.
> >
> > Granted that my concerns about error bars are met in the next revision, I have decided to raise my score.

---

> > > ### Author Response · Authors · 2021-08-20
> > > **We appreciate the positive decision**
> > >
> > > We are pleased to hear that our efforts are to the point and address Reviewer MbMP's concerns. We are committed to making the suggested improvements in the revision.

---

### Official Review · Reviewer_pbGK · 2021-07-17

**Rating:** 8
**Confidence:** 4

**Summary:**

The authors establish a relationship between the generalization error of trajectories obtained from a training algorithm and the persistent homology (PH) dimension. The theoretical contribution of this work involves combining two results: (1) box dimension of a bounded set can be computed using PH0 dimension [KLS06, Sch19], and (2) previous work linking box dimension and generalization error [SSDE20]. Based on these theoretical findings, the authors propose a simple algorithm to compute PH dimension directly, by performing a line fit on 0-dimensional topological features derived from weights in previous training iterations. This algorithm is based on mild assumptions, in contrast to previous work on computing the fractal dimension of training trajectories. Experiments applying this algorithm to a variety of networks (including AlexNet, CNNs and FCNs trained on MNIST, CIFAR10 and CIFAR100) and training algorithms (SGD, RMSprop and Adam) show that the PH dimension is inversely correlated with test accuracy. Next, the authors took advantage of the differentiability of persistent homology to incorporate the PH dimension computed from previous iterates as a regularizer to control the generalization error. The topological regularizer improved performance on the test dataset, especially at high learning rates where the unregularized network has low test accuracy. Finally, the authors performed ablation studies using synthetic data generated from β-stable Levy processes that exhibit heavy-tails observed in network trajectories. PH dimension outperformed other intrinsic dimension estimators in these tests, with varying number of points, ambient dimensions and line fitting procedures.


**Limitations And Societal Impact:**

Limitations are discussed in the manuscript. No negative societal impact expected from this work.

**Main Review:**

Originality:
This work combines previous theoretical results to propose a practical methodology for estimating intrinsic dimension using topological data analysis (TDA). Novel contributions include verification of the proposed PH dimension calculation algorithm on synthetic data and its practical application in multiple networks to demonstrate its usefulness in quantifying generalization error. The proposed regularization, based on PH dimension, is shown to be effective at controlling generalization error during training. All related work, datasets, and models are cited.
Quality:
The exposition is technically sound, and theoretical findings are supported by experiments. The authors provide context for their work within the broader scope of intrinsic dimension estimation, manifold reconstruction, bounds of generalization error, and the application of TDA to analyze deep networks. The authors clearly state that the α parameter should be smaller than the intrinsic dimension in theory, but the experiments were performed with fixed α = 1. I suggest the following improvements:
1) Definition 4: where PHi(VR(W)) is the i dimensional persistent homology of the Cech Vietoris-Rips complex on a finite point set …
2) Please consider including computational complexity and/or runtime performance of estimating PH dimension and the extra training time required when using PH dimension as a regularizer.
3) Figure 7 (main text) is hard to read and poorly labelled.
4) In SM Fig. 1, please indicate in the caption that PCA is outside the bounds of the axis limits.
5) In SM Fig. 7, consider using the same axis limits for all figures to allow for easy comparison. The observed changes in the persistence diagrams corresponding to changes in the intrinsic dimension are not explained. Why do points move closer to or away from the diagonal?
6) Similar to comment (3) above, the visualization of distance matrices (Fig. 4 in the main text, SM Figs. 5 and 6) showing non-uniform pixelation merits further explanation. How are the rows and columns of these matrices organized? Can the non-uniform pixelation be quantified and what does it represent?
Clarity:
The main paper and the supplementary material are very well organized and clearly written. I discovered only a few typos and grammatical mistakes that can easily be corrected. Reference to Fig. 5 in the supplement shows up as a ‘?’.
Significance:
The analysis of generalization error associated with various training algorithms is important for uncovering how deep networks learn and operate. This work is a significant contribution in this direction. The authors describe a practical algorithm, based on solid theoretical foundations, for estimating the intrinsic dimension of training trajectories and link it to generalization error. As demonstrated in this manuscript, the PH dimension can be used to evaluate different hyperparameters and control generalization loss during training. Higher-dimensional topological features computed from training trajectories may yield more insight in the future.

I am satisfied by the author responses and will maintain the high score.

**Time Spent Reviewing:**

6

---

> ### Author Response · Authors · 2021-08-10
> **Author Response to Reviewer pbGK**
>
> We thank the reviewer for their positive and encouraging feedback. We are happy that the reviewer found our paper technically sound and novel, our contributions to be significant. We thank the reviewer for acknowledging that our theoretical findings are supported by the experiments. We appreciate the suggested improvements and address the major ones below.
>
> *Please consider including computational complexity and/or runtime performance of estimating PH dimension and the extra training time required when using PH dimension as a regularizer.*
>
> For questions regarding computational complexity, we refer the reviewer to our “general response”.
>
> We note that, when using PH dimension as regularizer, one can control the extra training time by sacrificing dimension estimation accuracy by taking less samples and/or less points. For our experiments, we found that hyperparameter settings which resulted in an approximately 7x increase in time (8.9 seconds per epoch without regularization vs 62.5 seconds per epoch with regularization) produced the most consistent results with less overhead. We note that, as one increases the network size, this overhead would become less pronounced. Standard training would slow significantly while the only increase in speed for topology would be computing pairwise distances between higher dimensional points (which uses numpy vectorized computation). Furthermore, we stress that, compared with the highly optimized GPU computation for standard neural network operations, GPU computation of differentiable persistent homology is a nascent field with substantial room for improvement. We believe that further research along this line (which our current results motivate as a practical application) could lower the overhead required.
>
> We will include both computational complexity and training time in our final version of our paper.
>
> *In SM Fig. 7, consider using the same axis limits for all figures to allow for easy comparison.*
>
> Unfortunately setting the same axis limits would either make certain diagrams invisible or render a large portion of the information invisible. Moreover, we believe that, as much as the distance from the diagonal, what matters in this plot is the relative arrangement of the birth and death rates.
>
> *The observed changes in the persistence diagrams corresponding to changes in the intrinsic dimension are not explained. Why do points move closer to or away from the diagonal? Similar to comment (3) above, the visualization of distance matrices (Fig. 4 in the main text, SM Figs. 5 and 6) showing non-uniform pixelation merits further explanation. How are the rows and columns of these matrices organized? Can the non-uniform pixelation be quantified and what does it represent?*
>
> We agree with the reviewer that the semantics of the distance matrices were overlooked and we will include a more detailed discussion in the revision.
>
> The columns and rows of the distance matrices are organized with respect to the iteration indices after convergence: we first train the networks until convergence, and then run an additional 200 iterations near the local minimum. Then, the (i,j)th entry of a distance matrix corresponds to the Euclidean distance between the i-th iterate and the j-th iterate (of these additional 200 iterations).
>
> Though we do not yet have a rigorous proof, we believe that the qualitative difference in these diagrams is due to the **heavy-tailed** behavior of the SGD algorithm. Let us illustrate this point with a simpler example: consider the Levy $\beta$-stable process used in Section 5.2. This is a well-known heavy-tailed process which becomes heavier-tailed when the parameter $\beta$ decreases. A classical result in probability theory [A] shows that the Hausdorff dimension of the trajectories of this process is equal to its tail-index $\beta$ for **any** $d \geq 2$. This means that as the process gets heavier-tailed, its intrinsic dimension decreases.
>
> On the other hand, if we investigate the geometric properties of heavy-tailed processes, we mainly observe the following behavior: *the process behaves like a diffusion for some amount of time, then it makes a big jump, then it repeats this procedure*. For a visual illustration, we can recommend Figure 1 in [arxiv:2006.09313]. Finally, this “diffuse+jump” structure creates **clusters**, where each cluster is created during the “diffuse” period and a new cluster is initiated at every large “jump”.
>
> Now, coming back to the original question, the **non-uniform pixelations** in the distance matrices are well-known indications of clusters in data: in the top row of Figure 4, the large dark squares indicate different clusters, and we observe that these clusters become more prominent for smaller PH dimensions.
>
> To sum up:
> * 1- SGD can show heavy-tailed behavior when the learning-rate/batch-size is chosen appropriately [B,C].
> * 2- Heavy-tails result in a topology with smaller dimension [A], [SSDE20], and creates a clustering behavior in the trajectories [SSDE20].
> * 3- The clustering behavior results in non-uniform pixelations in the distance matrices and gaps in the persistence diagrams.
>
> We acknowledge that this connection might not be easily extracted from the current version of our paper. We will add a detailed explanation to the supplementary document.
>
> References:
>
> [A] Blumenthal, Robert M., and Ronald K. Getoor. "Some theorems on stable processes." Transactions of the American Mathematical Society 95.2 (1960): 263-273.
>
> [B] Hodgkinson, Liam, and Michael Mahoney. "Multiplicative noise and heavy tails in stochastic optimization." International Conference on Machine Learning. PMLR, 2021.
>
> [C] Gurbuzbalaban, Mert, Umut Simsekli, and Lingjiong Zhu. "The heavy-tail phenomenon in sgd." International Conference on Machine Learning. PMLR, 2021.
>
>
> *The authors clearly state that the α parameter should be smaller than the intrinsic dimension in theory, but the experiments were performed with fixed α = 1*
>
> The reviewer is totally right. We will clearly mention in the text that in our experiments we are assuming that the intrinsic dimension is larger than 1, hence we set $\alpha=1$. Nevertheless, we ablate the choice of $\alpha$ in Figure 7, and observe that $\alpha=1$ is a reasonable choice.

---

### Decision · Program_Chairs · 2021-09-27

**Decision:**

Accept (Poster)

**Comment:**

This paper identifies a notion of intrinsic dimension that can be rigorously linked to generalization performance. Given the scarceness of rigorous results in this field, it is an important achievement. It backs up the theoretical results by numerical experiments. A number of reviewers pointed out the difficulty to read and interpret the figures. It is important that the authors make on effort on that point before publication. The reviewers seem mostly convinced by the authors answers. After the rebuttal, reviewer J8S7 acknowledges the originality of the theoretical contribution of the paper (its main strength), but remains partially convinced by the answers concerning the large gap in terms of performance with respect to the state of the arts networks. I completely agree with reviewer J8S7 that such performance gap is a source of doubts. Therefore, I invite the authors to or discuss in details the reasons for that gap, and the choice behind these experiments despite this gap (that can be closed with simple networks), as they discuss in their response; or to improve the generalization. Otherwise, it gives an impression that the results of the paper may only apply in trivial regimes of (lack of) learning and therefore would not be relevant to applications. That being said, I have no doubts that all these improvements will be implemented by the authors and will only improve the overall quality.